# Metastatic melanoma: An integrated analysis to identify critical regulators associated with prognosis, pathogenesis and targeted therapies

Zeinab Chaharlashkar[1], Yousof Saeedi Honar[2], Meghdad Abdollahpour-Alitappeh[3], Sepideh Parvizpour[1,4]*, Abolfazl Barzegar[5]*, Effat Alizadeh[1]*

1 Department of Medical Biotechnology, Faculty of Advanced Medical Sciences, Tabriz University of Medical Sciences, Tabriz, Iran, 2 Department of Plant Biotechnology, Faculty of Life Sciences and Biotechnology, Shahid Beheshti University, Tehran, Iran, 3 Department of Physiology and Pharmacology, Pasteur Institute of Iran, Tehran, Iran, 4 Research Center for Pharmaceutical Nanotechnology, Biomedicine Institute, Tabriz University of Medical Sciences, Tabriz, Iran, 5 Department of Animal Biology, Faculty of Natural Sciences, University of Tabriz, Tabriz, Iran

* alizadehe@tbzmed.ac.ir, e.alizadeh.2010@gmail.com (EA); Barzegar@tabrizu.ac.ir (AB); se.parvizpour@gmail.com (SP)

**Data Availability Statement:** All relevant data are within the paper.

## Abstract

Metastatic melanoma causes a high rate of mortality. We conducted an integrated analysis to identify critical regulators associated with the prognosis, pathogenesis, and targeted therapies of metastatic-melanoma. A microarray dataset, GSE15605, including 12 metastatic-melanoma and sixteen normal skin (NS) samples, were obtained from the GEO database. After exploration of DEGs of NS and metastatic-melanoma, identification of relevant transcription factors (TFs) and kinases, the Gene Ontology (GO), and pathways analyses of DEGs were performed. Protein-protein interaction (PPI) networks were evaluated by the STRING and Cytoscape. Subsequently, the hub genes were selected using GEPIA. Survival analysis was performed using the TCGA. To identify microRNA and lncRNA DEGs of the melanoma-associated genes miRwalk and FANTOM6 were employed. In metastatic-melanoma samples 285 and 1173 genes were up and down-regulated, respectively. The upregulated genes were mostly involved in granulocyte chemotaxis, positive regulation of calcium ion transmembrane transport, and melanin biosynthetic process. Five hub genes including CXCL11, ICAM1, LEF1, MITF, and STAT1 were identified, SUZ12, SOX2, TCF3, NANOG, and SMAD4 were determined as the most significant TFs in metastatic-melanoma. Furthermore, CDK2, GSK3B, CSNK2A1, and CDK1 target the highest amounts of genes associated with disease. The DGIdb analysis results show the match drugs for five hub genes. MiRNAs analysis revealed hsa-miR-181c-5p, hsa-miR-30b-3p, hsa-miR-3680-3P, hsa-miR-4659a-3p, hsa-miR-4687-3P, and hsa-miR-6808-3P could regulate the hub genes, whereas RP11-553K8.5 and SRP14-AS1 were identified as the top significant lncRNA. The items recognized in the current study can be used as potential biomarkers for diagnostic, predictive, and might helpful to develop targeted combined therapies.

**Funding:** This study was supported by Tabriz University of Medical Sciences with grant number 69755. The funders had no role in study design, data collection and analysis, decision to publish, or preparation of the manuscript.

**Competing interests:** The authors have declared that no competing interests exist.

**Abbreviations:** GEO, Gene Expression Omnibus; DEGs, differentially expressed genes; TFs, transcription factors; GO, Gene Ontology; KEGG, Kyoto Encyclopedia of Genes and Genome; X2K, eXpression2Kinase; ChEA, ChIP enrichment analysis; PPI, Protein-protein interaction networks; GEPIA, Gene Expression Profiling Interactive Analysis; NS, Normal Skin.

# 1. Introduction

Melanoma cancer, derived from the malignant conversion of melanocytes, was reported to remain the most serious type of skin cancer, accounting for approximately 75% of derm cancer deaths [1, 2]. The prevalence of melanoma has risen quicker than other types of cancers over the past fifty years [3]. Melanoma cancer shows high metastatic spread with a median endurance of around 9 months, has a poor prognosis compared with primary melanoma and, importantly, is resistant to conventional treatment modalities, such as radiotherapy and chemotherapy [3–5]. In addition, patients with advanced/metastatic-melanoma almost suffer from an incurable disease, highlighting the urgent need for the seeking of more successful therapies [3, 4]. MicroRNAs, a taxonomy of non-coding RNAs playing key roles in tuning the expression of the genes and long non-coding RNAs (lncRNAs). The lncRNAs are lengthier than miRNAs about 200 nucleotides) were demonstrated to have functional roles in the initiation and continuation of various diseases, including melanoma. miRNA profiles of melanoma showed that different miRNAs regulate the translation or stability of the genes according to the disease stage, demonstrating their significant role in controlling events associated with melanoma. Importantly, a variety of studies have verified the important activity of lncRNAs as effective regulators in advanced malignant melanoma. Therefore, non-coding RNAs are eligible to be employed as potential biomarkers for diagnostic and prognostic metastatic-melanoma [5–9]. Similar to non-coding RNAs, transcription factors (TFs), kinase regulate genes, and cellular pathways take parts in the initiation and metastasis of different cancers [10–12], such as melanoma [13, 14]. With the fast improvement of high-throughput innovations, the role of bioinformatics analysis in identifying biomarkers of various diseases including cancer was highlighted [15]. The combination of bioinformatics and molecular biology can be used to identify molecular mechanisms and biomarkers for better cancer earlier detection and accurate treatments [16]. In the recent decade, bioinformatics-based approaches, such as Microarray analysis, have been used to discover and understand DEGs, pathways functioning in the DEGs, relations among proteins, and molecular mechanisms playing pivotal roles in the establishment and grow up of metastatic-melanoma [17–19]. The microarray technology has been also used to study other cancers [20]. Of note, understanding mechanisms, pathways, and markers involved in melanoma can be used as a target in clinical practice [9].

Here, we performed an analysis of the microarray data of metastatic-melanoma samples and NS samples, followed by determination of up-regulated versus down-regulated genes as metastatic-melanoma related or NS-correlated genes, respectively. Then, we analyzed GO, KEGG and PPI network construction and module identification, as well as survival analysis, miRNA and lncRNA analysis. Findings from our study demonstrated that some signaling pathways and key TFs, miRNAs, and hub genes are elaborate in regulatory metastatic-melanoma. The identified 10 hub genes, pathways and non-coding group among all RNAs can be used as potential novel targets in the control and diagnosis of metastatic-melanoma (Fig 1).

# 2. Methodology

## 2.1. Process and interpretation of microarray and gene expression data

The metastatic-melanoma reported gene expression findings were gained from NCBI located Gene Expression Omnibus (GEO) database (the URL is http://www.ncbi.nlm.nih.gov/geo). The gene expression profile dataset GSE15605 (GPL570 [HG-U133 Plus 2] Affymetrix Human Genome U133 Plus 2.0 Array) including 12 metastatic-melanoma samples and sixteen NS samples was retrieved from GEO. Utilizing GEO2r online tool, we normalized the higher and lower transcribed gene clusters between the metastatic-melanoma and NS samples.

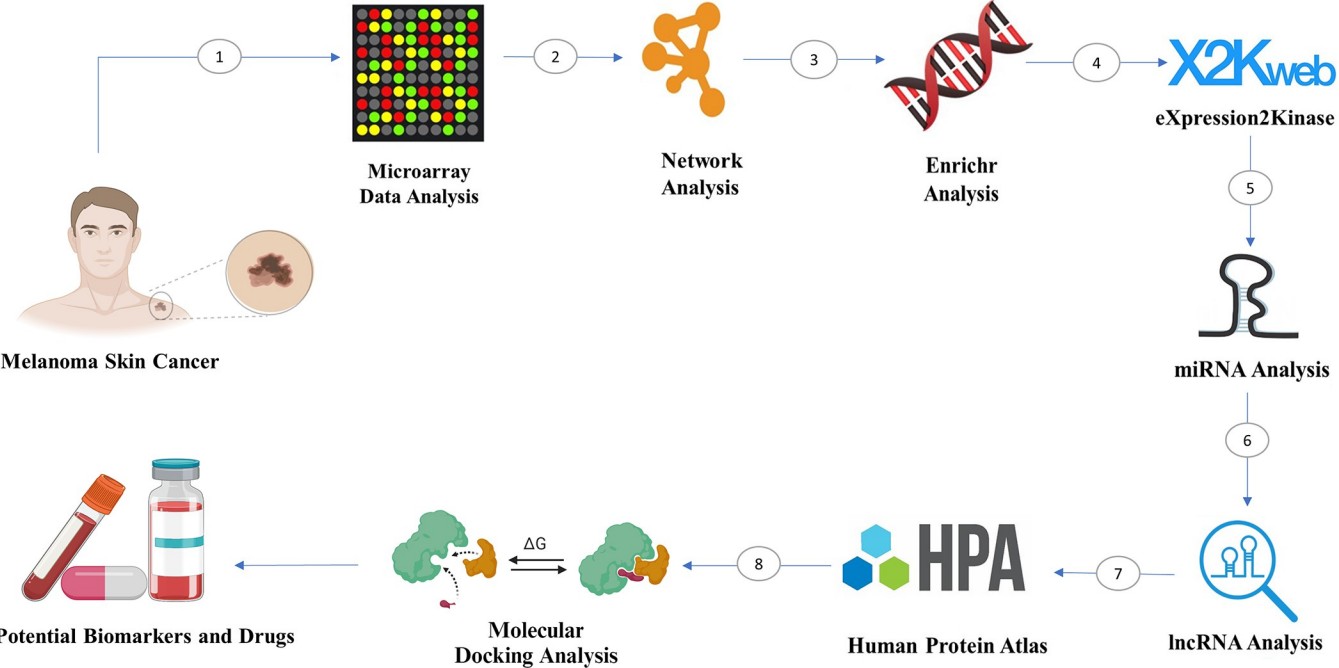

**Fig 1. The graphical abstract and schematic analyses carried out in the present study.**

Consequently, gene expression profiles were then compiled individually in an Excel file. Lastly, the *p* value of 0.05 was used to select gene clusters in this section. Ethics code for this study is: IR.TBZMED.VCR.REC.1401.131.

## 2.2. Gene set enrichment analysis

We used GO terms for functional enrichment analysis, which categorizes genes according to their molecular functions (MFs), biological phenomena (BPs), and cellular compositions (CCs). The KEGG databases were served as data sources for pathway enrichment analysis. A probability-value of equivalent/fewer than 0.05 was a significancy boarder in GO terms and pathways. One of the metastatic-melanoma analyzes performed for genes related to metastatic-melanoma is the use of the Enrichr database. Utilizing the Enrichr database, the functional enrichment analysis of DEGs was evaluated (https://maayanlab.cloud/Enrichr/).

## 2.3. PPI network and modular investigations

The STRING database (version 11.5; URL: https://string-db.org) was employed for predicting the possibly functional interactions amongst DEGs (A confidence score equal to 0.4 was considered as the cut-off standard for constructing PPI networks). Then, visualization of PPI networks was conducted using Cytoscape software (version 3.9.1;www.cytoscape.org). Therefore, Utilizing the Molecular Complex Detection (MCODE) plug-in of Cytoscape, the functional modules of the PPI networks were subsequently discovered (version 2.0.2), and lastly, Cyto-Hubba (version 0.1) plug-in the Cytoscape software were applied to find hub genes.

## 2.4. Hub gene selection and confirmation in the human protein atlas (HPA)

The HPA database (the URL is: https://www.proteinatlas.org/) was searched to recognize the DEGs hubs between patients and normal controls. This Swedish program was launched in

2003 in order to mapping entirely the human proteins in via the inclusion of several omics' technologies. We also visualized box plots generated by GEPIA (http://gepia.cancer-pku.cn/) to examine how the expression of hub genes differed between metastatic-melanoma tumors and normal tissues [21].

## 2.5. Hub genes mRNA expression validation

The mRNA expression status of the retrieved hub genes was confirmed using UALCAN (https://ualcan.path.uab.edu/index.html), an online resource that incorporates transcriptome sequencing data from TCGA [22]. To determine statistical significance in the survival analysis, a threshold of $P<0.05$ was applied. This approach lends credibility to the findings and can inform future investigation of these genes in the context of disease progression and prognosis.

## 2.6. Identification of kinases and TFs

The ChIP based enrichment analysis (ChEA) database was utilized to identify TFs that regulate the expression of metastatic-melanoma linked genes. The ChEA database contains information on eukaryotic type TFs, consensus binding sequences (positional weight matrices), experimentally verified binding sites, and controlled genes [23]. Additionally, eXpression2Kinases (X2K) (https://amp.pharm.mssm.edu/X2K/) was utilized to recognize and classify candidate TFs, protein kinases, and protein complexes that were probably responsible for the detected alterations in the metastatic-melanoma transcriptome [24].

## 2.7. MicroRNA (miRNA) target gene identification

The Enrichr dataset-connected miRTarBase (http://amp.pharm.mssm.edu) was employed to find the topmost miRNAs that potentially target metastatic melanoma connected certain genes considering $P \leq 0.05$.

## 2.8. LncRNAs target gene identification

LncRNAs may influence cell propagation, programmed cell death, and invasion during cancer growth. Therefore, we aimed to find lncRNAs involved in DEGs-related metastatic-melanoma genes. For this purpose, the FANTOM6 database (https://fantom.gsc.riken.jp/6/) was employed.

## 2.9. Drug-gene interaction prediction

To establish a drug-gene interaction network, researchers utilized DGIdb 3.0. This network illustrates connections between drugs and genes sourced from various resources. DGIdb 3.0 provides a convenient tool for searching and filtering to easily access preferred data. The hub genes with the most interaction with other proteins associated with metastatic-melanoma was submitted to DGIdb 3.0 to find the math FDA approved drugs.

## 2.10. Molecular docking study

The 3D structure of proteins from hub genes was obtained from the Protein Data Bank. Drug compound structures in SDF format were retrieved from the PubChem database. Molecular docking was carried out to predict how all the identified ligands bind to the target proteins. This docking procedure utilized the AutoDock Vina tool within the PyRx virtual screening software. PyRx is a tool used in computer-aided drug design to screen compound libraries against specific targets. The PyRx program's standard setup parameters were used for the docking procedure. The complexes having the most favorable binding energy (kcal/mol) were

selected for additional examination. Additionally, the interaction between the ligand and protein pair was displayed using the BIOVIA Discovery Studio Visualizer software.

## 2.11. Molecular dynamics simulation

The protein and drug complex structures were exposed to molecular dynamics (MD) simulation consuming the GROMACS 5.1.4 force field to study the stability of the binding. Throughout the simulation, intermolecular interactions were analyzed using GROMOS 96 as the suitable force field. The molecular environment was adjusted to a pH of 7.0 to determine the corresponding pKa values. Counter ions were introduced in adequate quantities to balance the charge of the complexes. The whole system underwent minimization with 400 steps using the steepest descent algorithm. Electrostatic interactions of the molecules were managed through the particle mesh Ewald method. Our simulations were directed at a temperature of 300 K for a period of 100 nanoseconds. The final structure obtained from the simulation path was employed to assess the protein geometry's accuracy and the folding of the structure. Alterations in the structures of the yielded complexes were monitored through the root-mean-square deviation (RMSD), root mean square fluctuation (RMSF), and the Solvent Accessible Surface Area (SASA). Additionally, the hydrogen bonds between the protein and ligand within the docked complexes were examined throughout the simulation.

## 2.12. MM/PBSA binding free energy (BFE) calculation

The binding energy calculations for the complexes were executed using the equation (Eq. 1): $\Delta Gbind = Grlc - (Grec + Glig)$, where Grlc denotes the free energy of the receptor-ligand complex, while Grec and Glig represent the unbound receptor and ligand, respectively. To assess the binding energies of the protein-ligand complexes, the g_mmpbsa tool was employed, which operates on the principles of the molecular mechanic/Poisson-Boltzmann surface area (MM/PBSA) method. Within this tool, the MM/PBSA method is utilized to compute different components of the binding energy, excluding the entropic term, and to assess each residue's energetic contribution to the binding using an energy decomposition approach. In this investigation, the free binding energies were computed based on the final 20 nanoseconds of the simulation. The MM/PBSA calculation for the protein-ligand complexes encompassed a total of 500 snapshots. Subsequently, the g_mmpbsa package was utilized to ascertain the residues' contributions to the binding energy.

## 3. Results

### 3.1. GO function and pathway enrichment examination of metastatic-melanoma and NS-related genes

The gene expression profile of GSE15605 between metastatic-melanoma (12 samples) and NS (16 samples) was estimated using the GEO2r, |log FC|≥2.5 and *P*-value<0.05. Our data discovered that 285 and 1173 genes were over and under expressed in metastatic-melanoma samples, respectively.

In this study, we identified the up and down regulation of genes in the status of metastatic-melanoma and in NS, respectively. According to GO analysis metastatic-melanoma-related genes were considerably enhanced in various proliferation related biology process (BPs), including granulocyte chemotaxis, positive regulation of calcium ion transmembrane transport and melanin biosynthetic process (Fig 2A), whereas NS samples related genes were particularly enriched in epidermis evolution, skin development and epidermal cell differentiation (Fig 2B). The analysis directed on cellular component (CC) displayed that the metastatic

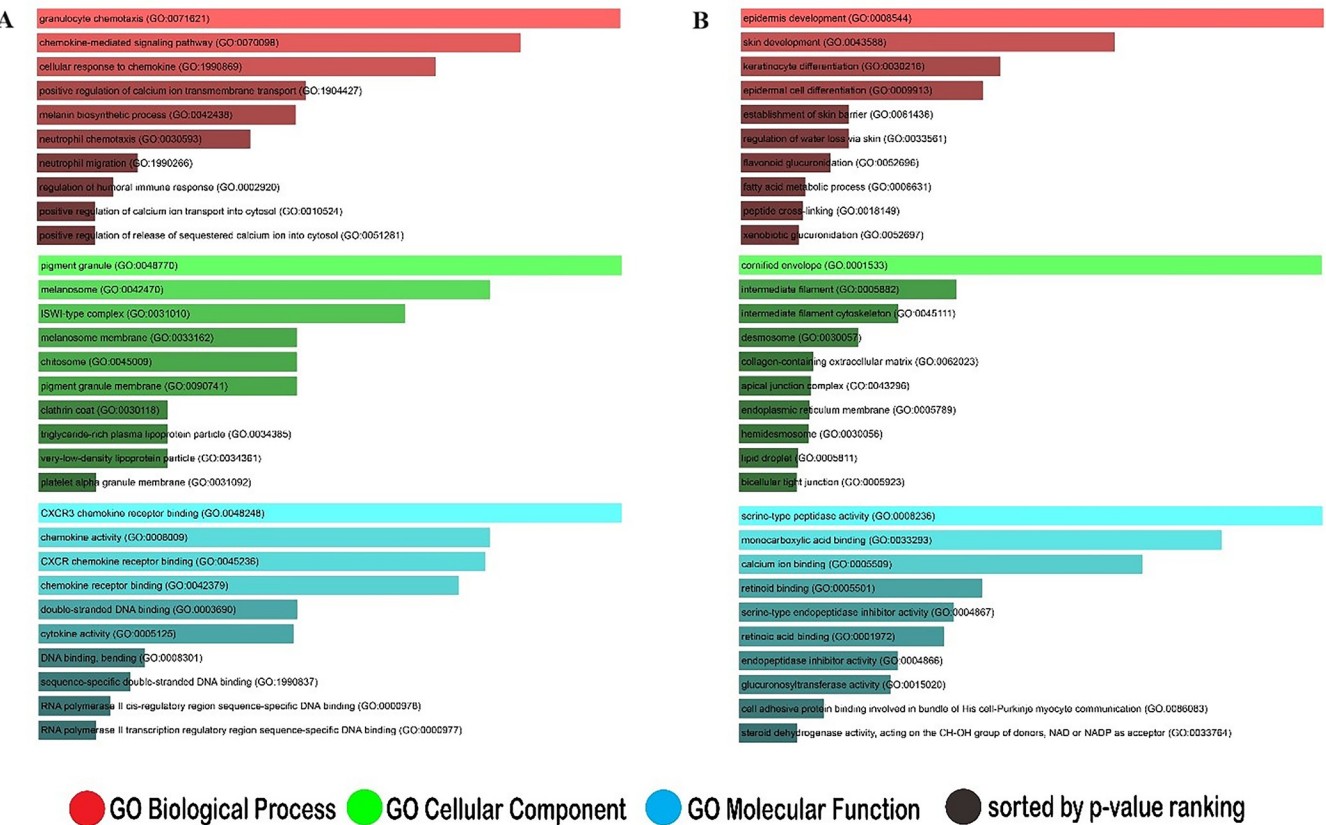

**Fig 2.** (A, B) Enrichment of gene sets associated with metastatic-melanoma and NS samples. (A) The top ten Gene Ontology (GO) terms associated with genes involved in metastatic-melanoma. (B) The top ten GO words connected to NS-related genes. BP (biological process), MF (molecular function), CC (cellular component).

melanoma-related genes were mostly enriched in pigment granule, melanosome, ISWI-type complex, melanosome membrane (Fig 2A), and the NS samples linked genes were accompanying with cornified envelope, intermediate filament and intermediate filament cytoskeleton (Fig 2B). In addition, MF analysis revealed that metastatic melanoma-related genes were meaningfully enriched in CXCR3 chemokine receptor binding, activity, and CXCR receptor binding (Fig 2A), while the NS samples associated genes were mostly involved in serine-type peptidase activity, monocarboxylic acid binding and calcium ion binding (Fig 2B). The KEGG analysis revealed that the metastatic melanoma-associated genes were significantly augmented in Toll-like receptor signaling pathway, chemokine signaling pathway, and viral proteins contact with cytokines and their corresponding receptors (Table 1).

### 3.2. PPI network construction and module identification

To identify the interactions between metastatic melanoma-related genes and NS samples-related genes, the STRING database was employed, and then their interactions were illustrated using Cytoscape as a PPI network [20]. Finally, a PPI network (150 nodes and 352 edges) for metastatic melanoma-related genes was analyzed using the cytohubba (nodes sorted by degree) plug-in for interactions, and 20 hub proteins with significant functions were discovered (Fig 3A). Among them, CD8A, SOX2, TYR, CXCL10 and ICAM1 had the most interaction with other proteins associated with metastatic melanoma. In addition, the gene list associated with

**Table 1. Top 10 KEGG pathway of the MM-related genes.**

| Term | P-Value | Overlap-Genes |
|---|---|---|
| Toll-like receptor signaling pathway | 3.53E-12 | [CXCL10, CXCL9, CXCL8, STAT1, CCL3, SPP1] |
| Chemokine signaling pathway | 1.88E-08 | [CXCL10, CXCL9, CXCL8, STAT1, CCL3] |
| Viral protein interaction with cytokine and cytokine receptor | 1.21E-07 | [CXCL10, CXCL9, CXCL8, CCL3] |
| Influenza A | 1.07E-06 | [CXCL10, CXCL8, STAT1, ICAM1] |
| Cytokine-cytokine receptor interaction | 9.08E-06 | [CXCL10, CXCL9, CXCL8, CCL3] |
| Rheumatoid arthritis | 1.14E-05 | [CXCL8, CCL3, ICAM1] |
| AGE-RAGE signaling pathway in diabetic complications | 1.42E-05 | [CXCL8, STAT1, ICAM1] |
| Kaposi sarcoma-associated herpesvirus infection | 1.01E-04 | [CXCL8, STAT1, ICAM1] |
| Epstein-Barr virus infection | 1.16E-04 | [CXCL10, STAT1, ICAM1] |
| Lipid and atherosclerosis | 1.39E-04 | [CXCL8, CCL3, ICAM1] |

NS samples was examined using the cytoHubba plugin, and the results (Fig 3B) indicated that KRT5, KRT14, LOR, FLG and EGFR are the key genes. Furthermore, based on the degree of importance, three significant modules were identified from the PPI network of metastatic melanoma-related genes: module 1 (14 nodes and 79 edges), module 2 (10 nodes and 40 edges) and module 3 (6 nodes and 14 edges) (Fig 3C). Lastly, the lists of hub genes of metastatic-melanoma-related genes and NS tasters related genes were compared, and the signature key genes of both groups were discovered (Fig 3D).

## 3.3. Hub gene selection and validation in the HPA

The important hub genes were determined examining PPI network of metastatic-melanoma-linked genes using cytohubba. Among the top most twenty genes we found to may associate with metastatic-melanoma-related genes, the 5 hub genes including CXCL11, ICAM1, LEF1, MITF and STAT1 were evaluated in the protein atlas server (Fig 4A). Furthermore, the GEPIA database was used to evaluate the expression of our candidate hub genes in metastatic-melanoma and NS samples. The results showed that the transcript frequency of hub genes is considerably higher in metastatic-melanoma samples than in NS samples (Fig 4B). The information of the protein atlas server includes a variety of items, such as the pathological section, which contains gene expression information. After hub gene evaluation, the information of five genes in skin cancer was evaluated.

## 3.4. Screening of hub genes by survival analysis

Using the UALCAN tool, we examined the association between hub gene expression and overall survival in melanoma and detected a substantial correlation among high levels of CXCL11, ICAM1 and STAT1 expression and poor prognosis (Fig 5).

## 3.5. The key TFs and kinases in metastatic-melanoma

ChEA was utilized to identify the TFs that may regulate the expression of metastatic-melanoma-associated genes. The investigation revealed that SUZ12, SOX2, TCF3, NANOG and SMAD4 are the maximum chief TFs that control the largest number of genes associated with metastatic-melanoma [25]. In addition, X2K was performed to identify the pivotal TFs, kinases and intermediate proteins active in the control of gene expression. The expression of genes linked to metastatic-melanoma. Among ten TFs, SUZ12 and EZH2 exhibited the higher frequency of interactions with middle proteins and corresponding kinases (Fig 6).

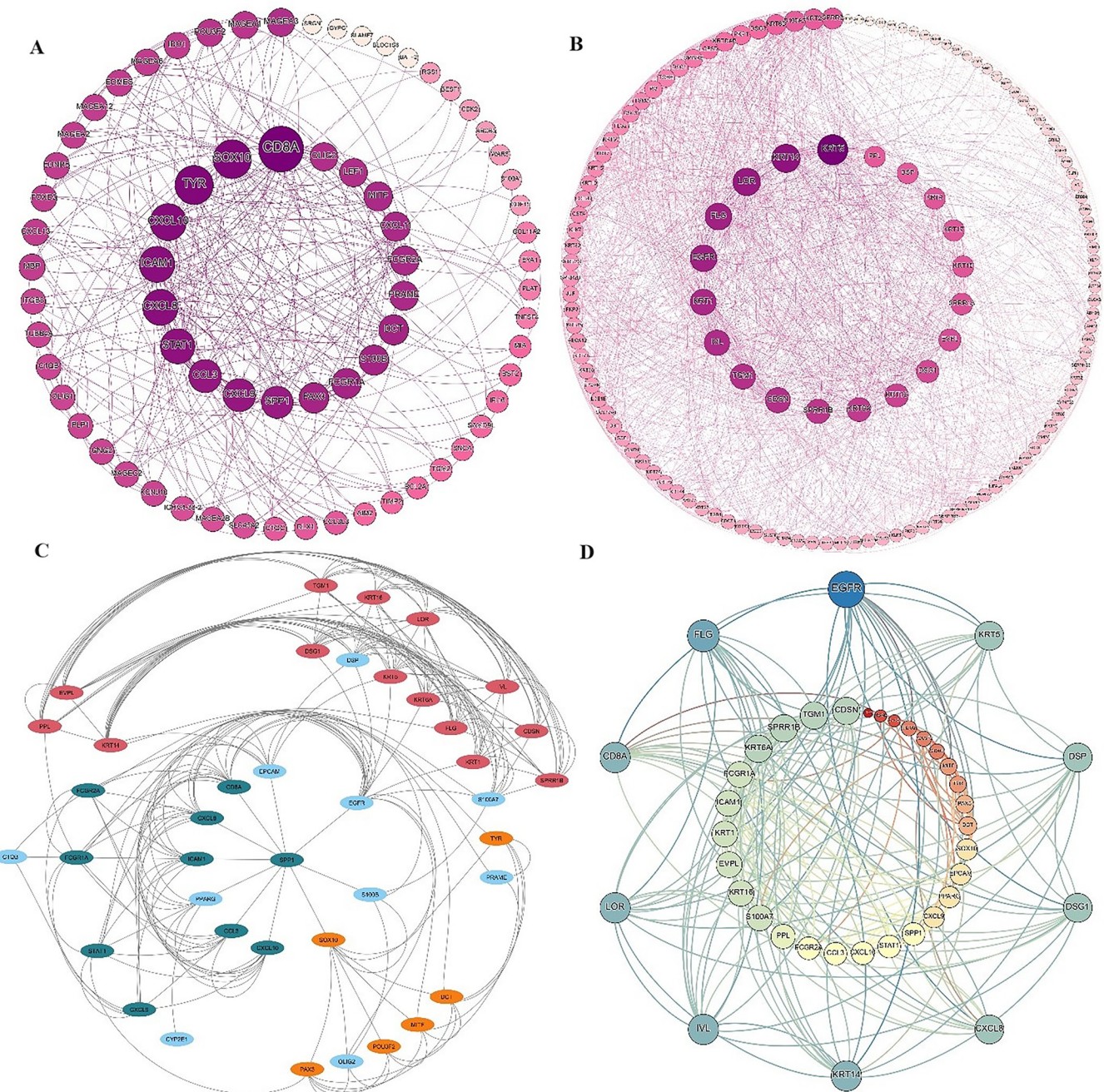

**Fig 3.** (A-D). The modules and networks of protein-protein interactions (PPIs). (A) PPI network of genes associated to metastatic melanoma. (B) PPI network of genes associated to NS. (C) Graphic illustration of the three most significant PPI network modules. (D) Analysis of Interaction between metastatic melanoma and NS samples related Hub Genes.

### 3.6. Identification of miRNAs targeting metastatic-melanoma-related genes

miRwalk was used to identify the miRNAs that may target metastatic-melanoma associated genes. The retrieved miRNAs were 152 that may target metastatic-melanoma-related genes have been identified. Then, for the top 10 hub genes, we found co metastatic-melanoma on microRNAs essential functions (Fig 6B), including hsa-miR-181c-5p, hsa-miR-30b-3p, hsa-

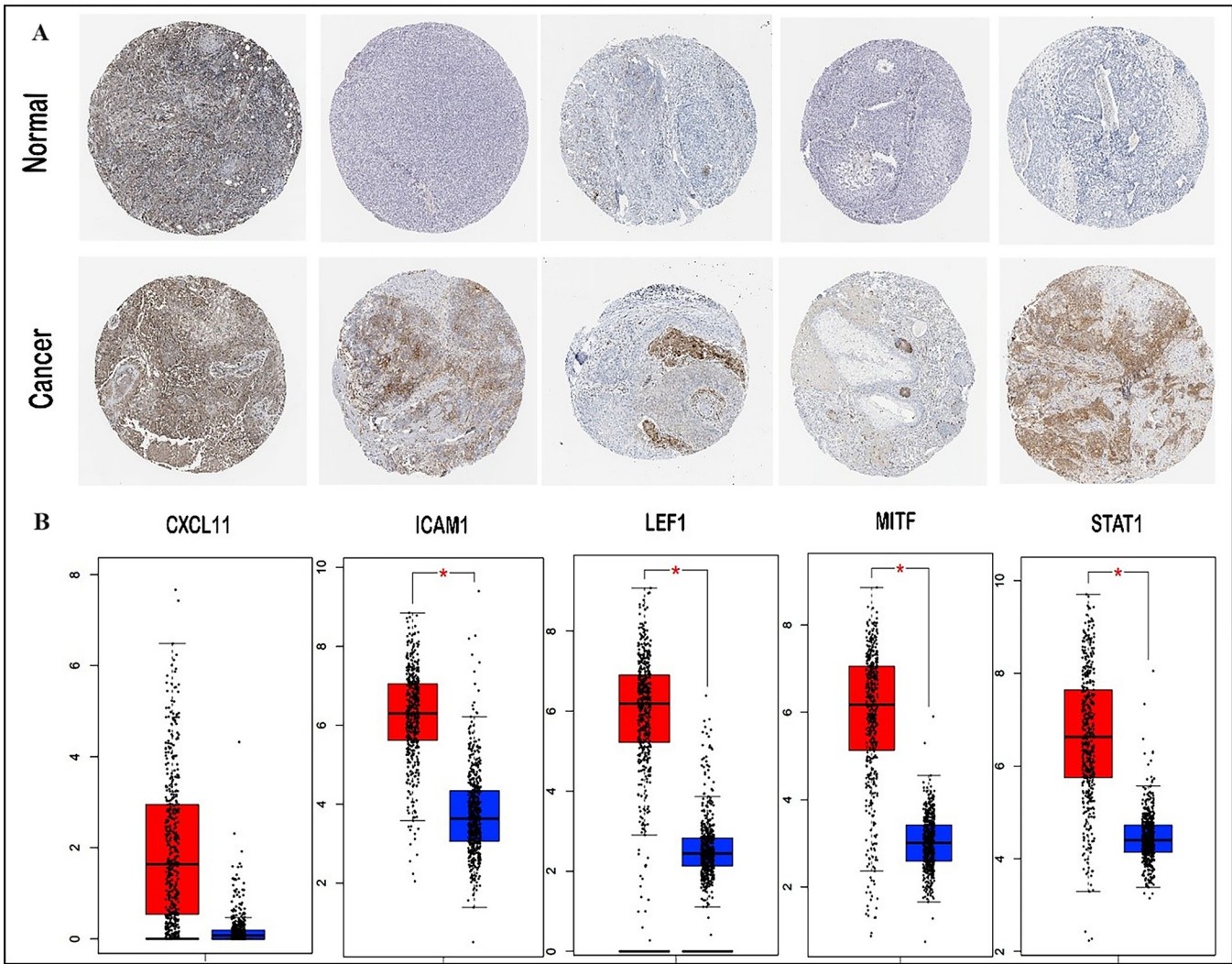

**Fig 4.** (A, B). Analyses of Protein Atlas. (A) Immunohistochemical examination of the five hub genes in metastatic melanoma (based on The Hub genes) [78] (Free License & Citation), (B) The expression levels of potential hub gene candidates (based on GEPIA database).

miR-6808-3p, hsa-miR-4687-3p, hsa-4659a-3p, and hsa-miR-3680-3p. For example, hsa-miR-181c-5p, only 6 targets were shared by the top 10 hub genes [26, 27] (Fig 7A).

### 3.7. LncRNAs that target metastatic-melanoma associated genes

LncRNAs are involved in a variety of hereditary disorders and cancer due to their key functions in gene regulation. They are a distinct group of RNAs that can be distinguished from other types based on their distinct properties, functions, and modes of action [28]. Using the FANTOM6 tool, we found top lncRNAs associated with metastatic-melanoma genes (Table 2).

### 3.8. The drug with gene interaction predictions

The DGIdb 3.0 database was utilized to explore potential medications for specific hub genes related to metastatic melanoma cancer and to extract information on drug-gene interactions.

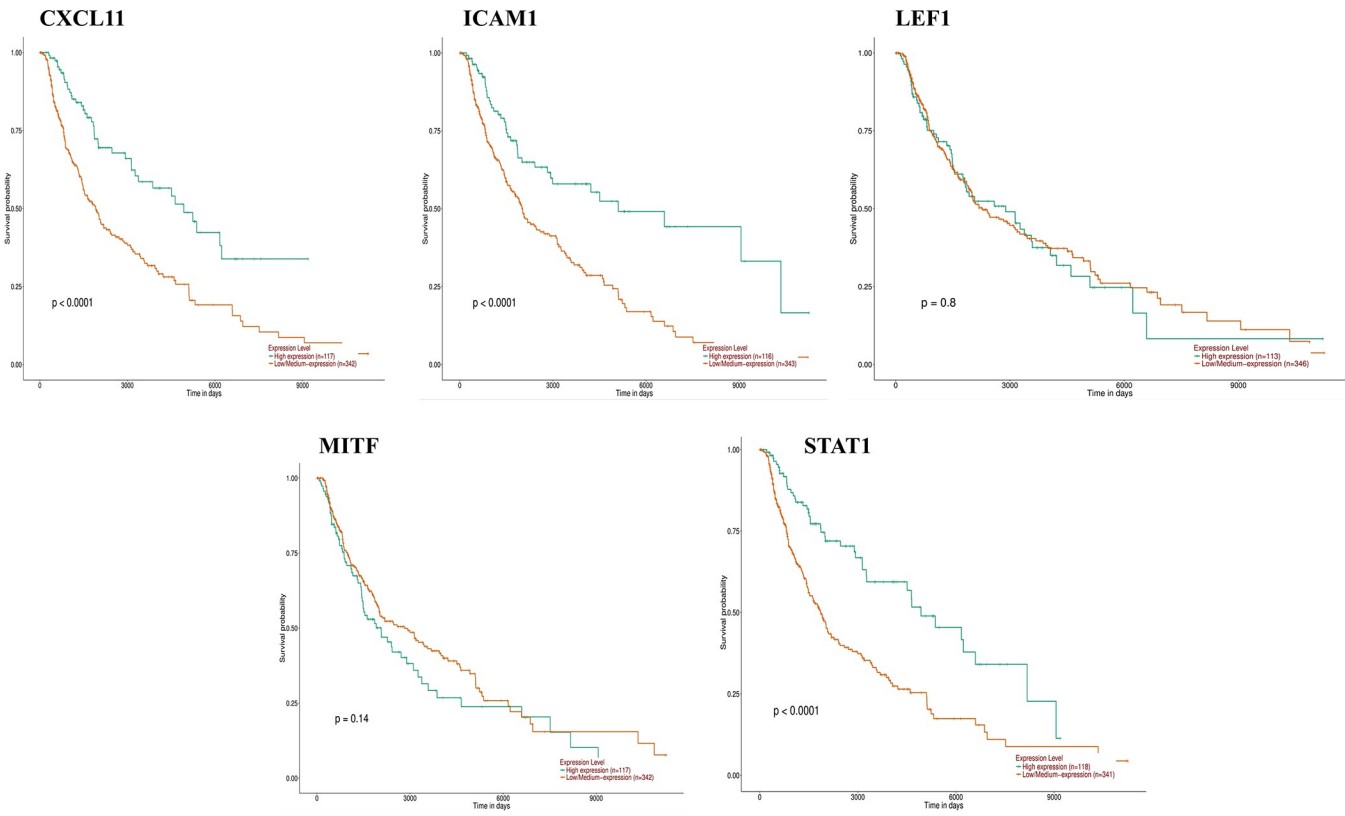

**Fig 5. Survival analysis of five hub genes in metastatic melanoma.**

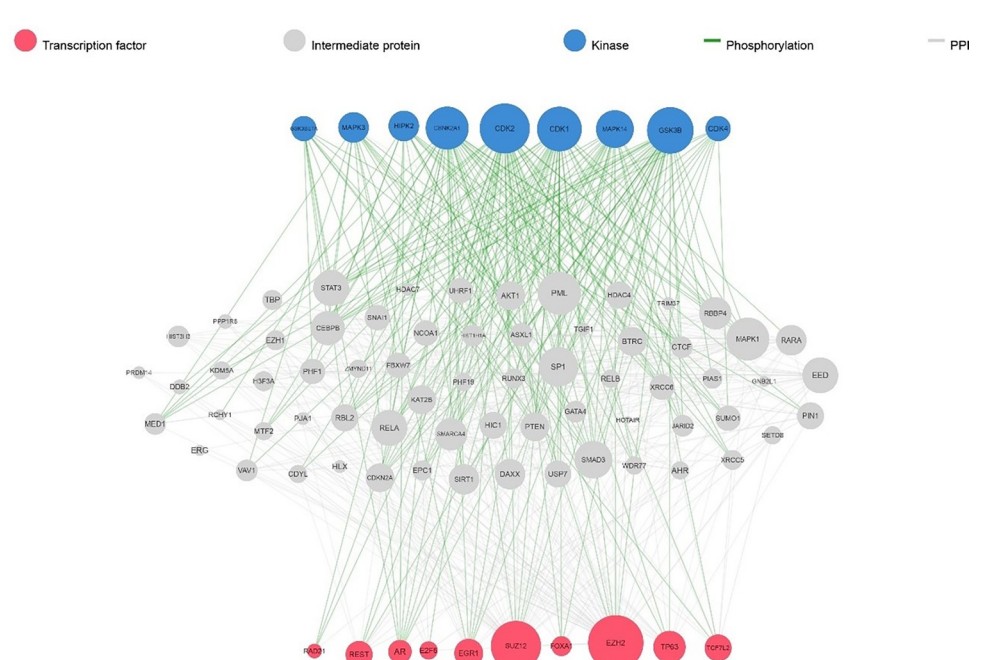

**Fig 6. Upstream regulatory network of the genes associated with metastatic-melanoma.** The color blue highlights the major kinases and the color red represents essential transcription factors (TFs).

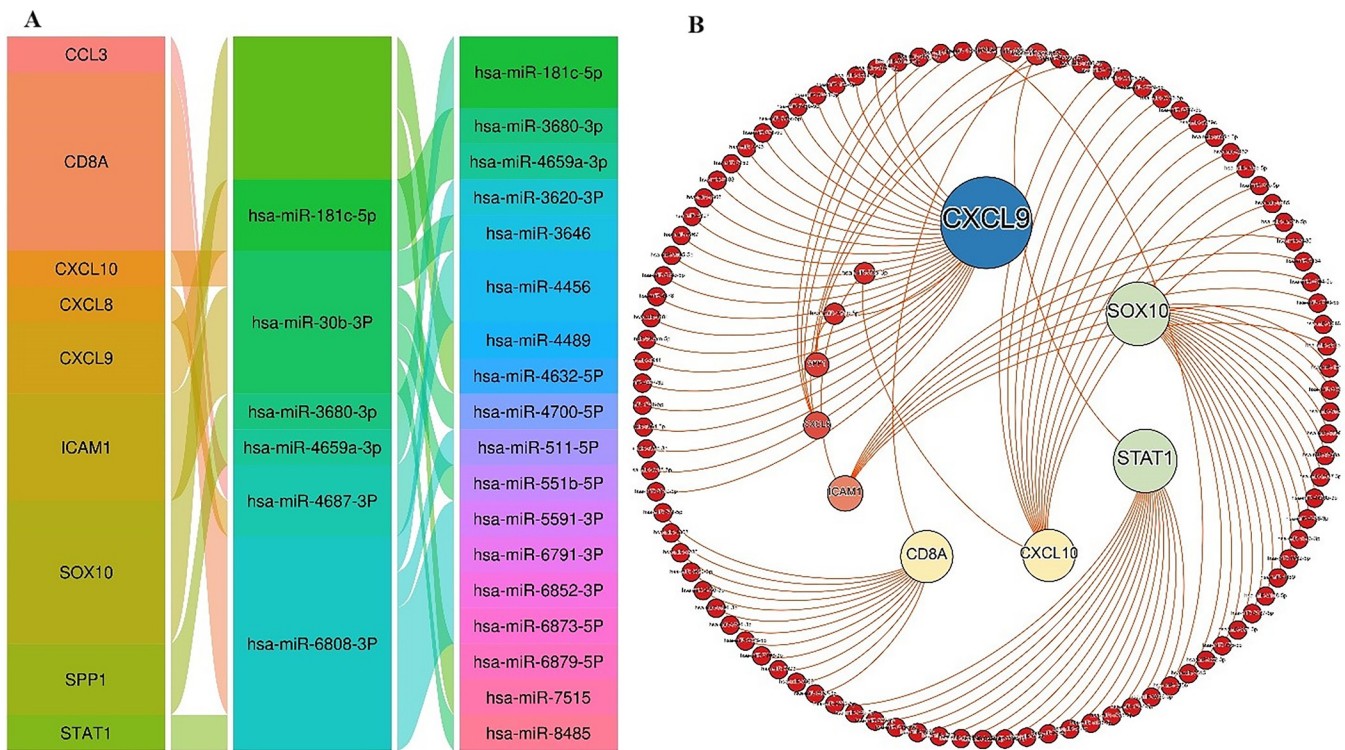

**Fig 7.** (A, B). Top ten microRNAs(miRNAs) that target metastatic-melanoma associated genes. (A) Table lists the common miRNAs whose function is to regulate the expression of the top up 10 genes. (B) illustrating the association between the top 10 genes and the individual and common miRNAs of each gene.

The identified drugs linked to the hub genes identified in the database are illustrated in Fig 8. The associations between the genes ICAM1, MITF and STAT1 and the drugs are presented separately in different groupings. In the visual representation, drugs are represented by pink nodes, while genes are represented by green nodes. Notably, no interactions were discovered for the ICAM1 and LEF1 genes.

## 3.9. Molecular docking analysis

The 3D structures of STAT1 (PDB ID: 1YVL), MITF (PDB ID: 4ATI), and ICAM1 (PDB ID: 1P53) were achieved from the PDB database. The drug structures identified in the DGIDB database were retrieved from the PubChem database. Molecular docking between the drug

**Table 2. Top significant *p*-values for FANTOM6 lncRNA KD DEGs of the MM-related genes.**

| Term | P-Value | Overlap-Genes |
|---|---|---|
| RP11-553K8.5-ASO_G0261573_AD_04-DEGs Up | 0.002493 | [CXCL8, STAT1, SPP1, ICAM1] |
| SRP14-AS1-ASO_G0248508_10-DEGs Down | 0.003609 | [CXCL8, ICAM1] |
| C11orf95-ASO_G0188070_AD_05-DEGs Down | 0.013915 | [ICAM1] |
| RP13-20L14.6-ASO_G0265458_AD_03-DEGs Up | 0.017091 | [CXCL8, SPP1, ICAM1] |
| LINC00862-ASO_G0203721_05-DEGs Up | 0.036881 | [CXCL8] |
| LL22NC03-86G7.1-ASO_G0224086_04-DEGs Up | 0.037365 | [SPP1] |
| RP5-1103G7.4-ASO_G0225377_05-DEGs Up | 0.038887 | [STAT1, SPP1, ICAM1] |

MM: metastatic-melanoma

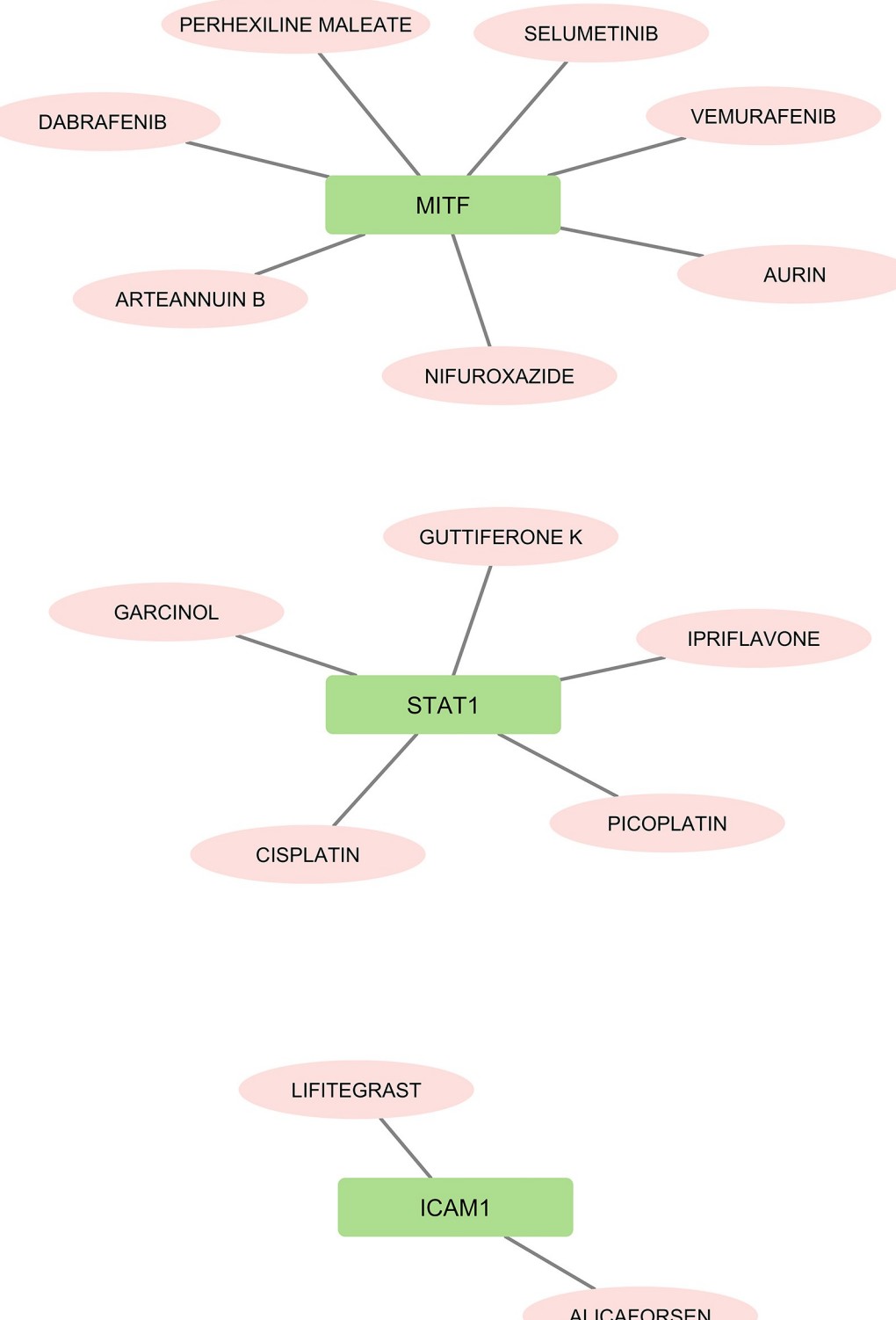

**Fig 8. The relationships between the, ICAM1 MITF, and STAT1 genes and drugs are shown in independently in three groups.** The pink and green nodes represent drugs and genes, correspondingly.

**Table 3. The binding affinities (kg/mol) of each docking interaction between proteins and drugs.**

| Protein | Drug | Drug | Drug | Drug | Drug | Drug | Drug |
|---|---|---|---|---|---|---|---|
| STAT1 | Cisplatin | Picoplatin | Garcinol | Ipriflavone | Guttiferone k | | |
| Binding Affinity (kg/mol) | -5.9 | -5.5 | -7.4 | -7.5 | -8.9 | | |
| MITF | Perhexiline maleate | Nifuroxazide | Aurin | Selumetinib | Arteannuin B | Vemurafenib | Dabrafenib |
| -Binding Affinity (kg/mol) | -3.4 | -5.7 | -7.9 | -5.9 | -5.0 | -6.0 | -8.1 |
| ICAM1 | Lifitegrast | Alicaforsen | | | | | |
| Binding Affinity (kg/mol) | -8.9 | -6.8 | | | | | |

compounds from the DGIDB database and each target protein (STAT1, MITF, ICAM1) was conducted using the AutoDock Vina wizard within PyRx. The binding affinities of each docking interaction are presented in Table 3.

### 3.10. Molecular dynamics simulation analysis

The structural stability and flexibility of the three complexes were evaluated via MD simulation at 300 K for 100 nanoseconds. Fig 9 illustrates the RMSD plot of the complexes during the simulation, indicating that all complexes maintained stable throughout the simulation period. Additionally, in Fig 9, the RMSD graphs of the STAT1-Guttiferone k, MITF-Dabrafenib and ICAM1-Lifitegrast complexes are depicted in blue, yellow, and green, respectively. The RMSF is primarily used to characterize and analyze local conformational changes within a protein structure. The Fig 10, the RMSF graphs of the STAT1-Guttiferone k, MITF-Dabrafenib, and ICAM1-Lifitegrast complexes are depicted in green, blue, and orange, respectively. Furthermore, the Fig 11 illustrates the SASA plot of all three complex structures within the simulation. The result shows that the MITF-Dabrafenib complex has higher SASA alteration than other two. The protein complexed with the selected ligands and their intermolecular interactions were analyzed using the Simulation Interactions Diagram (SID) during the 100 ns simulation run. These interactions (or "contacts") for the compounds with STAT1-Guttiferone k, MITF-Dabrafenib, and ICAM1-Lifitegrast complexes have been described based on hydrogen bonds, hydrophobic interactions, ionic interactions, and water bridges, as shown in Fig 12.

### 3.11. MM/PBSA BFE calculation

The final BFE in terms of the three complexes was computed to assess the energy contribution of non-covalent interactions. Different energy components influencing protein-ligand binding, such as van der Waals (Evdw), electrostatic (Eelec) and solvation (Esolvation) energies, were calculated to examine the overall BFE and complex stability. The MM/PBSA method for endpoint free-energy simulation was employed to delve deeper into the BFE of the complexes. Table 4 indicates the MM/PBSA results.

## 4. Discussion

Metastatic-melanoma is a really concerning cancer type that presents significant challenges for effective treatment due to the multipart molecular mechanisms work together and the resistance of tumors to current therapies [19]. Hence, exploring and understanding the molecular

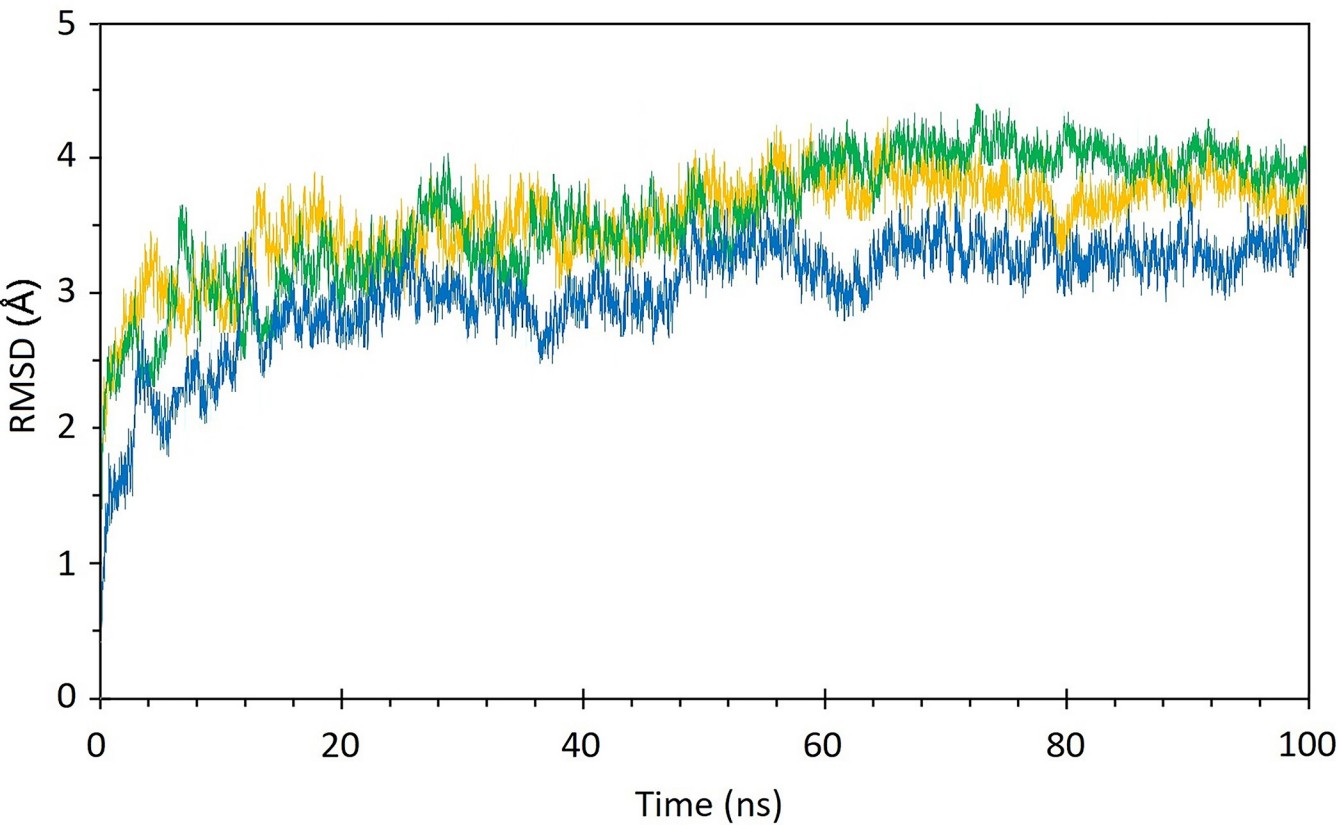

**Fig 9. The RMSD graphs of STAT1-Guttiferone k, MITF- Dabrafenib and ICAM1-Lifitegrast complexes are shown in blue, yellow, and green respectively.**

pathways, mechanisms, and biomarkers that regulate metastatic-melanoma may result in notable progressions in therapeutic approaches and broaden the range of clinical options for treating patients with metastatic-melanoma. In this work, the gene expression profile of GSE15605 to identify genes associated with metastatic-melanoma was subjected to analysis. Our findings revealed that 285 genes were in up-regulated status, while 1173 genes were detected in very low transcript numbers in metastatic-melanoma samples. We classified the over-expressed and under expressed genes as being related to metastatic-melanoma and NS samples, respectively. We conducted several bioinformatics analyses to identify signaling pathways, GO terms, hub genes, miRNAs, and lncRNAs that play critical roles in the spread of advanced melanoma. Our GO analysis disclosed that the upregulated DEGs were predominantly associated with granulocyte chemotaxis, positive regulation of calcium ion transmembrane transport, and melanin biosynthetic process. The upregulated DEGs were found to be mostly enriched in several signaling pathways linked to cancer, according to outcomes of KEGG pathway analysis. These pathways include chemokine signaling pathway, the Toll-like receptor signaling pathway, and viral protein interaction with cytokine and receptors of cytokines.

Our data was in line with previous reports that the Toll-like receptor signaling pathway is elaborated in the innate immune system and performs a serious duty in the recognition of pathogens [29]. The dysregulation of this pathway can lead to an impaired immune response to tumors, which can contribute to cancer development and progression [30]. The chemokine signaling pathway is active in the migration and activation of immune cells, which play a major character in tumor progression and metastasis [31]. Dysregulation of the chemokine

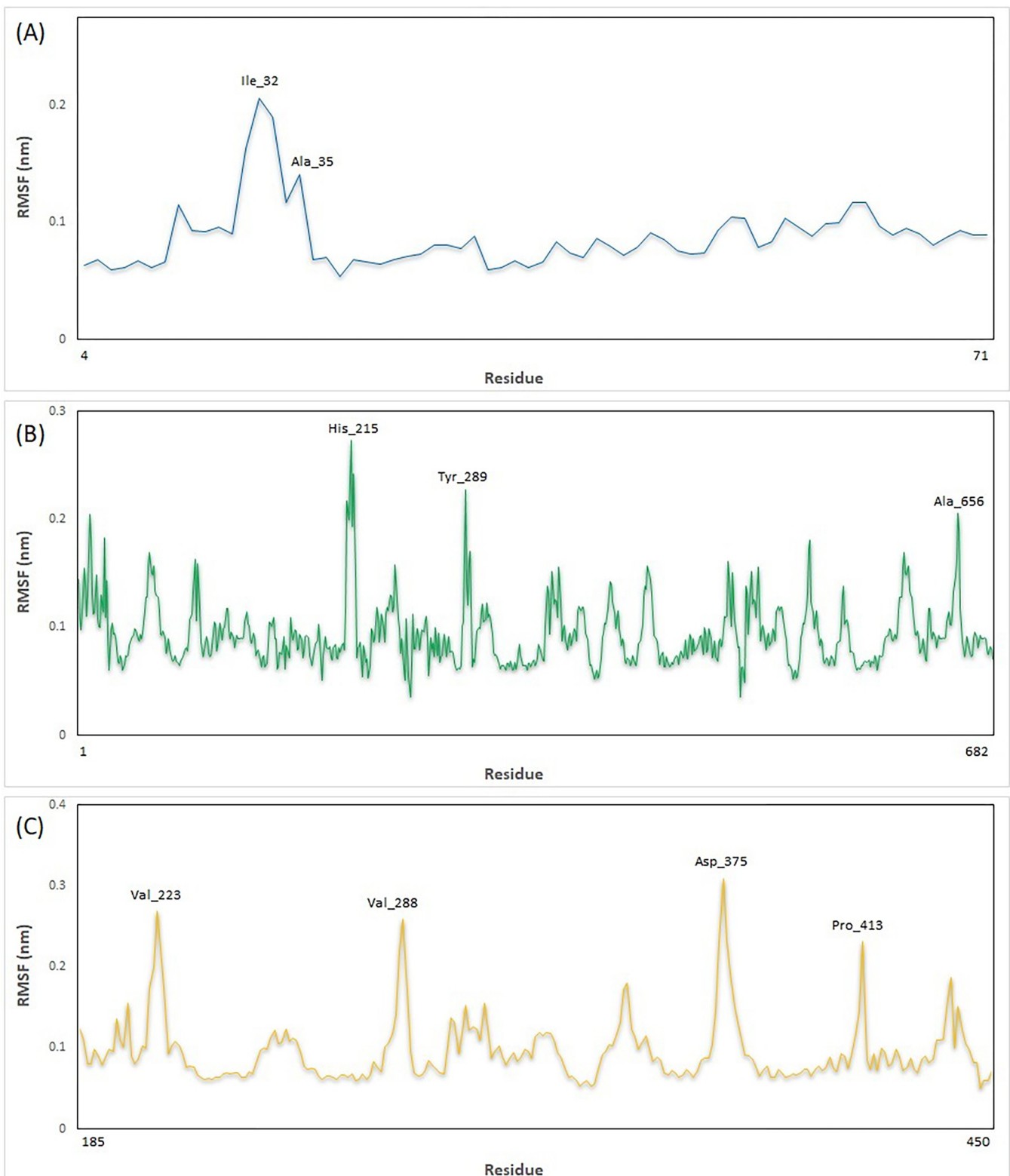

**Fig 10. The RMSF graphs of the STAT1-Guttiferone k, MITF-Dabrafenib, and ICAM1-Lifitegrast complexes are depicted in green, blue, and orange, respectively.**

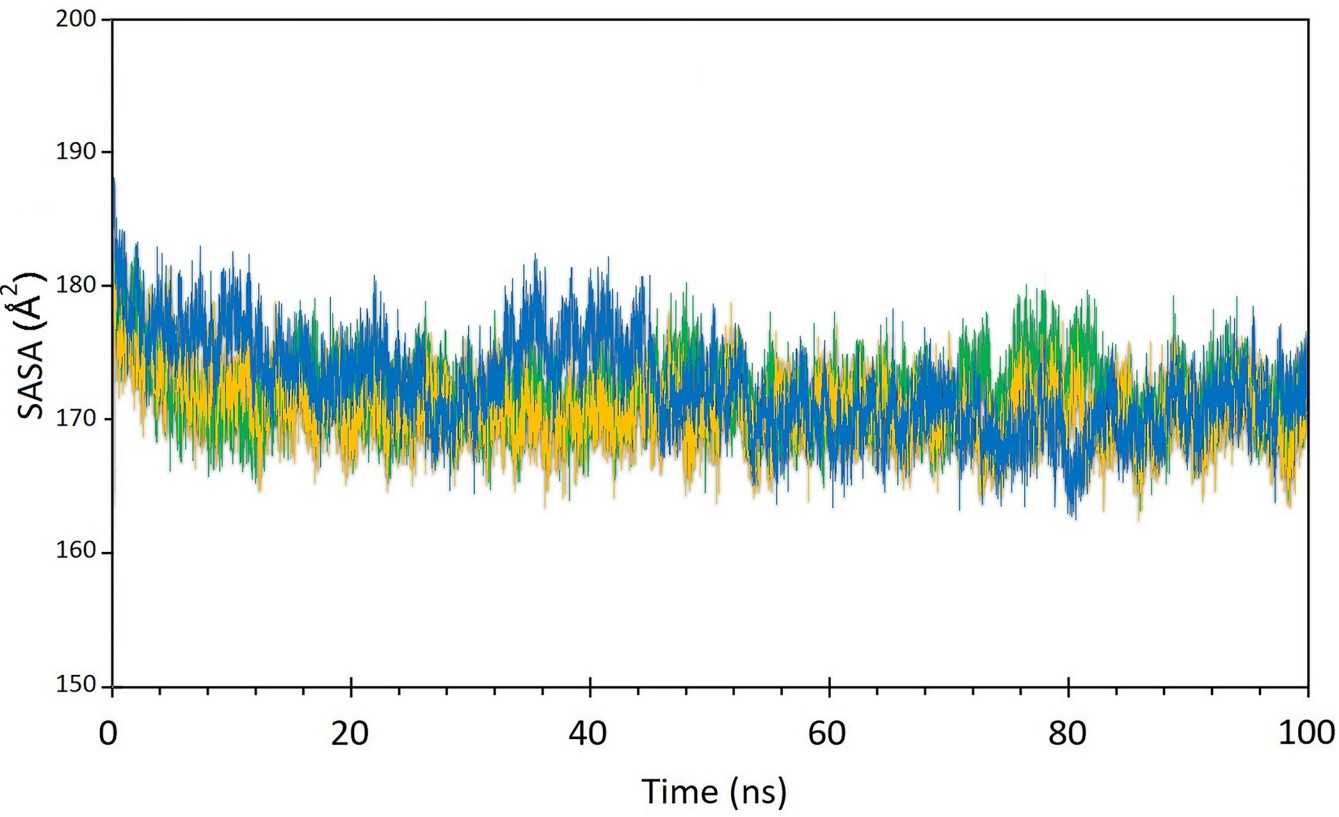

**Fig 11. The solvent accessible surface area (SASA) graphs of the STAT1-Guttiferone k, MITF-Dabrafenib, and ICAM1-Lifitegrast complexes are depicted in green, blue, and orange, respectively.**

signaling pathway has been connected with numerous types of cancer, including breast, prostate, and ovarian cancer. In cancer, this pathway can contribute to the recruitment of immune cells to the tumor niche, which can trigger tumor enlarge and attack [32].

Dysregulation of cytokine and cytokine receptors pathway has been implicated in various cancers, for example liver and cervical cancer [33]. Overall, the enrichment of upregulated DEGs in these signaling pathways suggests their potential involvement in cancer development and progression.

The PPI network was created to detect interactions among DEGs in cancer metastasis. The top five up-regulated DEGs in the network were CXCL11, ICAM1, LEF1, MITF, and STAT1. To validate these hub genes, their mRNA and protein expression levels were analyzed via the GEPIA and The HPA databases, respectively [21, 34]. Survival analysis also disclosed that the differential expression of CXCL11, ICAM1, and STAT1 was accompanying through poor prognosis in progressive melanoma.

CXCL11 is among the protein members of the CXC chemokine family, and its induction through interferon (IFN) suggests its functional involvement in the evolution of various cancers. It plays a vital role in stimulating anti-tumor activity mediated by immune cells and serves as a prognostic biological marker in colon adenocarcinoma (COAD). A number of studies have shown that CXCL11 modulates the immune responses in many cancers, for instance melanoma, prostate, liver, breast, stomach, and colorectal cancer. Its overexpression has been associated with a rise in T lymphocytes and a higher survival rate in patients [35].

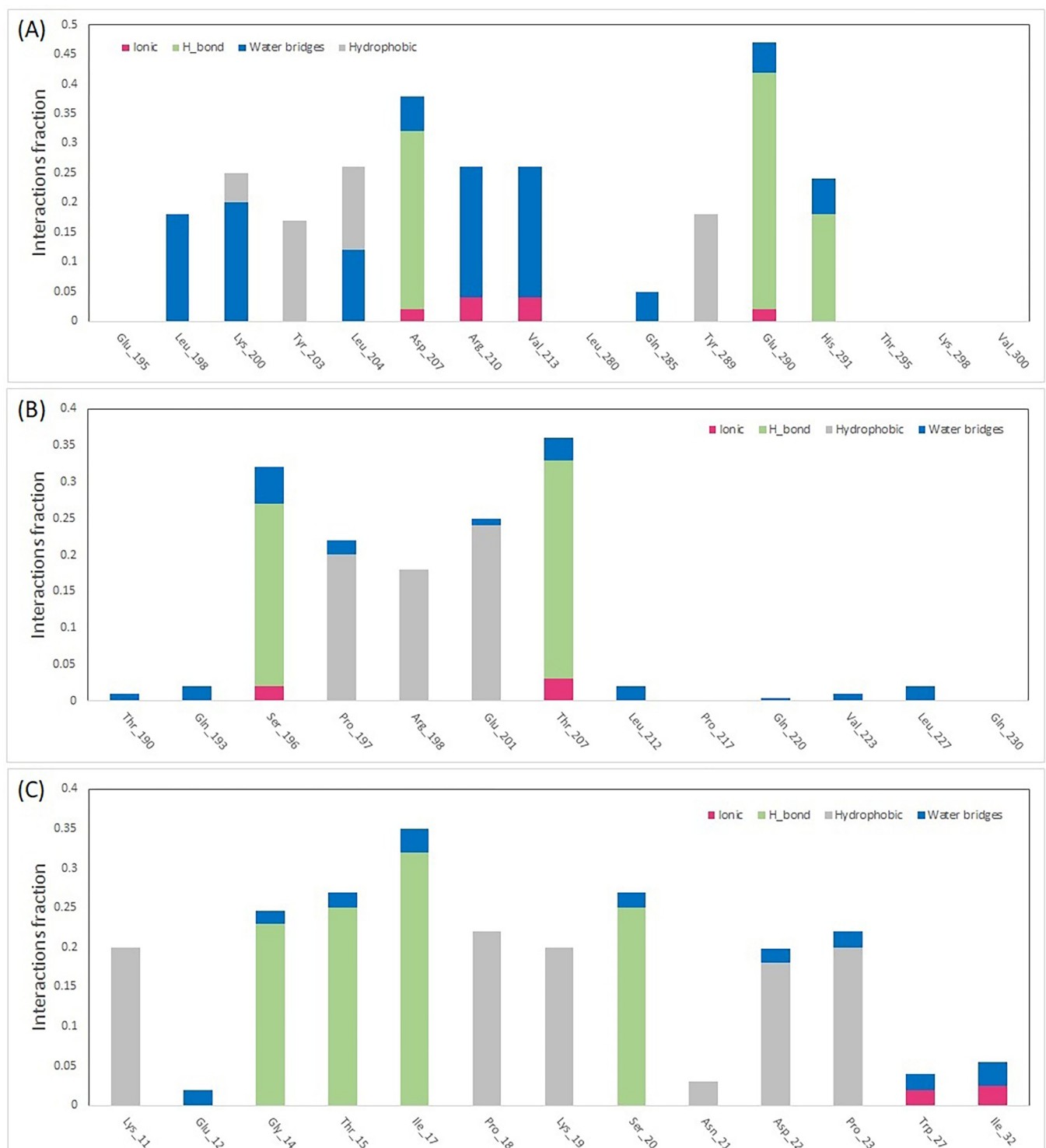

**Fig 12. The protein–ligand contacts during 100ns MD simulation.** (**A**) STAT1-Guttiferone k, (**B**) ICAM1-Lifitegrast, and (**C**) MITF-Dabrafenib.

**Table 4. The binding free energy (BFE) plot using molecular mechanic/Poisson–Boltzmann surface area for the three complexes STAT1-Guttiferone k, MITF- Dabrafenib and ICAM1-Lifitegrast.** Various energy components are illustrated, including Van der Waals (ΔEvdw), electrostatic energy (ΔEElec), polar solvation energy (ΔEPol), apolar solvation energy (ΔEAPool), and total binding energy (ΔEBinding).

| Compound | van der Waals Forces (kJ/mol) | Electrostatic Energy (kJ/mol) | Polar Solvation Energy (kJ/mol) | Apolar solvation energy (kJ/mol) | Binding Energy (kJ/mol) |
|---|---|---|---|---|---|
| STAT1-Guttiferone k | -189.49 | -29.91 | 90.55 | -17.20 | -119.06 |
| MITF- Dabrafenib | -170.89 | -19.13 | 87.05 | -17.68 | -94.64 |
| ICAM1-Lifitegrast | -200.80 | -35.62 | 120.48 | -18.05 | -131.99 |

Intercellular adhesion molecules (ICAM1) are members of the immunoglobulin superfamily. ICAM1 takes a part in the metastasis of some cancers and declines the immune response in cancer cells. Its inhabitation has also reduced invasion and migration in breast, lung, liver, colon, and thyroid cancer [36–39]. ICAM1 involves in the stimulation of immune response by promoting the adhesion and migration of leukocytes to the site of inflammation or infection. It accomplishes this through its interaction with the integrin receptors on leukocytes, which allows for firm attachment and transmigration of the leukocytes across the endothelial cell layer and into the tissue [40]. In cancer, ICAM1 overexpression has been proved in several types of tumors, including melanoma, breast, colorectal, and, lung cancer [39]. Its upregulation can support tumor development and metastatic spread by enhancing tumor cell adhesion, invasion, and migration. In addition, ICAMs, lead to enhanced mass tumor, angiogenesis, epithelial-mesenchymal transition (EMT), and immunosuppressive TME via STAT -3 activation. Numerous studies have documented that increased expression of L1CAM in melanoma cells inhibits tumor suppressors including p53 and PTEN, which frequently reported to become down-regulated in melanoma, permitting these cells to escape cell death. Therefore, inhibition of L1CAM represents another promising target against melanoma [41].

Lymphoid Enhancer-Binding Factor 1 (LEF1), has a critical role in the Wnt/β-catenin signaling pathway, which is involved in controlling several cellular functions, including tissue homeostasis, embryonic development, and cancer progression [42]. Inhibition of LEF1 expression with sodium selenite, Paclitaxel (PTX), 5-aza-2'-deoxycytidine and ethacrynic acid (EA) has provided potential therapy for cancer treatment [43–45]. Consequently, targeting LEF1 appears to be beneficial in suppressing cancer proliferation. Two of the last hub genes, MITF and STAT1 are involved in human cancers. Several investigations have discovered that the changing expressions of these genes lead to Melanoma cancers [46–49]. In this research, ChEA and X2K algorithms were utilized to identify possible TFs and kinases that might regulate the transcriptional levels of genes actively involved in metastatic-melanoma. The findings of the study revealed that SUZ12, SOX2, TCF3, NANOG, and SMAD4 were significantly linked with the expression of hub genes. We found that, compared to other transcription factors, SUZ12 has been found to have a more significant impact on regulating the expression of genes associated with metastatic-melanoma. This highlights the potential importance of targeting SUZ12 as a therapeutic strategy in the treatment of melanoma [50]. The DGIdb 3.0 database was utilized to explore potential medications for five detected hub genes. The associations between the genes ICAM1, MITF, and STAT1 and the drugs are presented in Fig 8. Notably, no interactions were discovered for the ICAM1 and LEF1 genes.

A molecular docking investigation was directed to identify the most favorable interactions between the drugs listed in DGIdb 3.0 and the 3 target proteins. The docking analysis revealed that the drug compounds interacted with all three target proteins, with interactions featuring lower BEF indicating increased complex stability. Subsequently, an MD simulation was carried out to evaluate the stability of the drug candidates when interacting with the target proteins.

The MD simulation outcomes indicated that all 3 complexes maintain stable in their interactions to the drug candidates, as illustrated in Fig 9. Notably, the energy profiles of all complexes consistently displayed low values. The MD simulation outcomes indicated that all 3 complexes maintain stable in their interactions to the drug candidates, as illustrated in Fig 9. Also, the RMSD graph shows that the MITF-Dabrafenib complex has lower value than others that indicates its more stability during simulation. The RMSF is primarily used to characterize and analyze local conformational changes within a protein structure. The Fig 10, the RMSF graphs of the STAT1-Guttiferone k, MITF-Dabrafenib, and ICAM1-Lifitegrast complexes are depicted in green, blue, and orange, respectively. Furthermore, the Fig 11 illustrates the SASA plot of all three complex structures within the simulation. The result shows that the MITF-Dabrafenib complex has higher SASA alteration than other two. The protein-ligand complexes were further examined for their MM/PBSA binding energy, as depicted in Table 4. Notably, the energy profiles of all complexes consistently displayed low values. Moreover, MITF-Dabrafenib complex shows lower energy than two other complexes. In next step, the protein complexed with the selected ligands and their intermolecular interactions were analyzed using the Simulation Interactions Diagram (SID) during the 100 ns simulation run. These interactions (or "contacts") for the compounds with STAT1-Guttiferone k, MITF-Dabrafenib, and ICAM1-Lifitegrast complexes have been described based on hydrogen bonds, hydrophobic interactions, ionic interactions, and water bridges, as shown in Fig 12. The summary of MD simulation results show that all three drug protein complexes are stable during simulation and can be suggest for future work, however, among them MITF-Dabrafenib complex is more stable.

Additionally, our examination showed that CDK2, GSK3B, CSNK2A1, and CDK1 were the significant kinases that target the greatest number of genes associated with metastatic-melanoma. CDK2 regulates G2 progression, G1 to S transition, and DNA synthesis. The abnormal CDK2 activation results in uncontrolled cell growth and cell division in various human cancers [51]. CDK2 induces resistance to BRAF and hsp90 inhibitors and a high ratio of CDK1 expression leads to resistance of melanoma patients to therapeutic agents which causes a significant decrease in OS. The CDK2 inhibition reduced resistance to both kinds of inhibitors [52]. Moreover, the knockdown of CDK2 through the CRISP/Cas9 approach induced apoptosis in cutaneous melanoma cells [53]. These results show that targeting CDK2 may improve clinical outcomes in melanoma patients. Abnormal GSK3B expression and action contribute to the progression of many types of cancer [54–57]. GSK3B facilitates tumor cell maintenance, spread, tumor invasion, and resistance to therapies [58]. Several studies also demonstrated that the suppression of GSK3B function decreased tumor cell survival, proliferation and promoted apoptosis in osteosarcoma, pancreatic, gastrointestinal, and glioblastoma cancer cells as well as other aggressive tumors such as urogenital, colon carcinoma, and lung cancer [59–66]. This evidence strongly validates that GSK3B is an significant target in cancer therapy. CSNK2A1 (also known as CK2α1) is a greatly conserved kinase involved in a variety of biological and pathological events. Overexpression of CSNK2A1 was found in advanced melanoma. Zhou et al. showed that high expression of CSNK2A1 leads to increased resistance to RAF-MEK kinase inhibitors (trametinib, vemurafenib, and dabrafenib) by sustaining ERK phosphorylation in BRAF-mutated melanoma cells. On the contrary, Knockdown of CSNK2A1 increases cell sensitivity to kinase inhibitors [67]. Moreover, inhibition of CSNK2A1 in combination with BRAF inhibitors has a lethal synergistic effect by reducing AKT signaling in melanoma and thyroid cancer cells [68]. CDK1, or cyclin-dependent kinase, is involved in the control of cell proliferation and cycle [69], and similar to CDK2, overexpression of CDK1 has been revealed in multiple malignancies, such as metastatic-melanoma. High CDK1 expression was also associated a lower overall viability [70]. Zhu et al. stated that an elevated expression level of CDK1 correlates with poor clinical outcomes in colorectal cancer and that inhibiting CDK1

increases 5-fluorouracil sensitivity index in patients suffering from colorectal cancer [71]. Furthermore, it has been demonstrated that the CDK1/Sox2 axis acts as an important axis in the control and maintenance of the stemness of lung cancer cells, and inhibiting CDK1 increases the sensitivity of lung cancer to chemotherapeutics [72]. Therefore, targeting these kinases may be an effective modality to control metastatic-melanoma. MicroRNAs (miRNAs) performs a vital duty in the epigenetic control of cellular biological approaches, such as cancer metastasis. Fluctuations in miRNA levels can lead to emergence of several forms of cancers [73]. The miRNA analysis with miRwalk revealed miRNAs that could actually target metastatic-melanoma- related genes such as hsa-miR-181c-5p, hsa-miR-30b-3p, hsa-miR-3680-3P, hsa-miR-4659a-3p, hsa-miR-4687-3P, hsa-miR-6808-3P. These miRNAs almost have a tumor inhibitor and mediate apoptosis activity. Recently, it was discovered that downregulation of hsa-miR-181c-5p increases cell migratory behavior or metastasis in Laryngeal squamous cell carcinoma (LSCC) [74]. Some earlier studies showed that hsa-miR-4659a-3p, and hsa-miR-4687-3P are closely related to tumor progression and metastatic potential in cancers [75, 76]. Instead, the sensitivity of tumor cells to chemotherapeutic medications increases through upregulation of these miRNAs. MiR-181c sensitizes chronic myelocytic leukemia (CML) to Adriamycin by targeting ST8SIA4 expression [77]. Taken together, these miRNAs might be used as a new tumor marker for the early discovery of metastatic-melanoma.

## 5. Conclusion

Overall, our study highlights the importance of utilizing innovative approaches and cutting-edge technologies to unlock innovative insights into the progress and treatment of advanced melanoma. We are hopeful that our findings will motivate performing upcoming research and novelty in this field, eventually leading to better-quality outcomes for melanoma cancer patients.

## Author Contributions

**Conceptualization:** Effat Alizadeh.

**Data curation:** Zeinab Chaharlashkar, Yousof Saeedi Honar, Meghdad Abdollahpour-Alitappeh, Sepideh Parvizpour, Abolfazl Barzegar, Effat Alizadeh.

**Formal analysis:** Zeinab Chaharlashkar, Yousof Saeedi Honar, Meghdad Abdollahpour-Alitappeh, Sepideh Parvizpour, Abolfazl Barzegar, Effat Alizadeh.

**Investigation:** Zeinab Chaharlashkar, Yousof Saeedi Honar, Sepideh Parvizpour, Abolfazl Barzegar, Effat Alizadeh.

**Methodology:** Zeinab Chaharlashkar, Yousof Saeedi Honar, Sepideh Parvizpour, Abolfazl Barzegar, Effat Alizadeh.

**Project administration:** Effat Alizadeh.

**Resources:** Zeinab Chaharlashkar, Yousof Saeedi Honar, Meghdad Abdollahpour-Alitappeh, Sepideh Parvizpour, Abolfazl Barzegar, Effat Alizadeh.

**Software:** Zeinab Chaharlashkar, Yousof Saeedi Honar, Meghdad Abdollahpour-Alitappeh, Sepideh Parvizpour, Abolfazl Barzegar, Effat Alizadeh.

**Supervision:** Effat Alizadeh.

**Validation:** Meghdad Abdollahpour-Alitappeh, Effat Alizadeh.

**Visualization:** Meghdad Abdollahpour-Alitappeh, Effat Alizadeh.

**Writing – original draft:** Zeinab Chaharlashkar, Yousof Saeedi Honar, Sepideh Parvizpour, Abolfazl Barzegar, Effat Alizadeh.

**Writing – review & editing:** Effat Alizadeh.

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
