## [Decision Letter · Decision Letter 0]

21 Aug 2024

PONE-D-24-19761Metastatic melanoma: An integrated analysis to identify critical regulators associated with prognosis, pathogenesis and targeted therapies

PLOS ONE

Dear Dr. Alizadeh,

Thank you for submitting your manuscript to PLOS ONE. After careful consideration, we feel that it has merit but does not fully meet PLOS ONE’s publication criteria as it currently stands. Therefore, we invite you to submit a revised version of the manuscript that addresses the points raised during the review process.

1. The manuscript does not seem to have conducted any non-computational experiments (or computational experiments to analyze the cohort from the author's institution). At the same time, GSE15605 is a relatively small data sample (12+16), which makes the generalizability of the results may not be proven. The author should consider explaining this part or adding necessary data/experiments to provide necessary evidence for generalizability.

2. Regarding the statement "The items recognized in the current study can be used as potential biomarkers for diagnostic, predictive, and might helpful to develop targeted combined therapies" mentioned in the manuscript. The experimental results do not seem to properly support the evidence mentioned in the manuscript. From my professional perspective, the manuscript seems to use virtual docking + molecular dynamics methods for analysis. As for molecular dynamics data, the relevant data of RMSF and SASA seem to be missing. As for the MMPBSA method, Figure 9 is very confusing, because such data may be more convincing in the form of a table. In addition, molecular dynamics does not seem to give a "conclusion". In addition, for example, Dabrafenib and MITF. In 2014, the FDA has approved Dabrafenib for use in patients with unresectable or metastatic melanoma with BRAF V600E mutations. To a certain extent, this coincides with the manuscript. In this context, is the addition of MITF a poly pharmacological effect? Or is it just a "computational coincidence"? More importantly, will it provide a proven potential predictor for more precise targeted therapy? Compared with a mechanical view of the compound, these issues should be discussed or anticipated by the author.

3. I personally suggest that the manuscript should add a graphic Abstract to attract readers. And many of the manuscript's expression systems face major revisions (for example, some pictures are marked as Fig, and some are Figure).

We look forward to receiving your revised manuscript.

Kind regards,

Marianne Clemence, Staff Editor, on behalf of,

Ruo Wang

Academic Editor

PLOS ONE

https://doi.org/10.1007/s12079-021-00612-8

In your revision ensure you cite all your sources (including your own works), and quote or rephrase any duplicated text outside the methods section. Further consideration is dependent on these concerns being addressed.

4. Funding Information and Financial Disclosure sections do not match:

We note that the grant information you provided in the ‘Funding Information’ and ‘Financial Disclosure’ sections do not match. 

[None]. 

6. Thank you for stating the following financial disclosure: 

 [The authors would like to acknowledge the Department of Medical Biotechnology, Faculty of Advanced Medical Sciences, Tabriz University of Medical Sciences for all support provided [69755].].  

7. Thank you for stating the following in the Acknowledgments Section of your manuscript: 

[The authors would like to acknowledge the Department of Medical Biotechnology, Faculty of Advanced Medical Sciences, Tabriz University of Medical Sciences for all support provided [69755]]

  [The authors would like to acknowledge the Department of Medical Biotechnology, Faculty of Advanced Medical Sciences, Tabriz University of Medical Sciences for all support provided [69755].]. 

8. In the online submission form, you indicated that [All data will be presented upon request.]. 

9. Your ethics statement should only appear in the Methods section of your manuscript. If your ethics statement is written in any section besides the Methods, please move it to the Methods section and delete it from any other section. Please ensure that your ethics statement is included in your manuscript, as the ethics statement entered into the online submission form will not be published alongside your manuscript. 

10. We note that Figure(s) 3a  in your submission contain copyrighted images. All PLOS content is published under the Creative Commons Attribution License (CC BY 4.0), which means that the manuscript, images, and Supporting Information files will be freely available online, and any third party is permitted to access, download, copy, distribute, and use these materials in any way, even commercially, with proper attribution. For more information, see our copyright guidelines: http://journals.plos.org/plosone/s/licenses-and-copyright.

. You may seek permission from the original copyright holder of Figure(s) 3a to publish the content specifically under the CC BY 4.0 license. 

b If you are unable to obtain permission from the original copyright holder to publish these figures under the CC BY 4.0 license or if the copyright holder’s requirements are incompatible with the CC BY 4.0 license, please either i) remove the figure or ii) supply a replacement figure that complies with the CC BY 4.0 license. Please check copyright information on all replacement figures and update the figure caption with source information. If applicable, please specify in the figure caption text when a figure is similar but not identical to the original image and is therefore for illustrative purposes only.

11. Please include a copy of Table 1, 2 and 3 which you refer to in your text on page 11 and 13.

Additional Editor Comments:

The manuscript was highly praised by the peer reviewers. Congratulations : )

Reviewers' comments:

Reviewer's Responses to Questions

**Comments to the Author**

1. Is the manuscript technically sound, and do the data support the conclusions?

Reviewer #1: Yes

Reviewer #2: Yes

Reviewer #3: Yes

2. Has the statistical analysis been performed appropriately and rigorously? 

Reviewer #1: Yes

Reviewer #2: Yes

Reviewer #3: Yes

3. Have the authors made all data underlying the findings in their manuscript fully available?

Reviewer #1: Yes

Reviewer #2: Yes

Reviewer #3: Yes

4. Is the manuscript presented in an intelligible fashion and written in standard English?

Reviewer #1: Yes

Reviewer #2: Yes

Reviewer #3: Yes

5. Review Comments to the Author

Reviewer #1: The manuscript entitled "

Metastatic melanoma: An integrated analysis to identify critical regulators associated

with prognosis, pathogenesis and targeted therapies" is interesting ,althought the manuscript lacks experimental data , but bioinformatic and statistical analysis are robust.

Reviewer #2: This is nice piece of research work. They have done a great job to identify genes or factors responsible for metastatic melanoma cancer. This is a dangerous disease for the human being. So, this research has a great impact on human health concerning skin disease.

Comment 1: Its ok. It will be great to confirm the function of the identified genes on metastatic melanoma cancer in biological manner as it is so important for disease treatment.

Reviewer #3: Paper titled Metastatic melanoma: An integrated analysis to identify critical regulators associated with prognosis, pathogenesis and targeted therapies are well documented and can beAccept in the present form

6. PLOS authors have the option to publish the peer review history of their article (what does this mean?). If published, this will include your full peer review and any attached files.

Reviewer #1: **Yes: **Shahram Teimourian

Reviewer #2: **Yes: **Dr. Md Sohel Hasan, Professor, Dept. of Biochemistry and Molecular Biology, Rajshahi University, Bangladesh

Reviewer #3: **Yes: **Dr. Saumya Patel Gujarat University

---

## [Author Response · Author response to Decision Letter 0]

9 Oct 2024

Date: 09/29/2024

Manuscript ID: PONE-D-24-19761

Submission of revise

Dear Professor Editor/s,

Thank you very much for giving us the opportunity to submit a revised version of our manuscript entitled “Metastatic melanoma: An integrated analysis to identify critical regulators associated with prognosis, pathogenesis and targeted therapies” for publication in the esteemed Journal Plos One. We appreciate the time and efforts that you and the reviewers dedicated to provide feedback on our manuscript and are grateful for the insightful comments on and valuable improvements to our paper. We incorporated all of the suggestions made by the reviewers and editors. Please see below, in blue, for a point-by-point response to the reviewers’ comments and concerns. Also, we uploaded a track change version including comments within the revised manuscript as well as a clean revised manuscript. Thank you in advance. We look forward to hearing from you. 

Sincerely,

Dr. Effat Alizadeh 

Associate Professor, Department of Medical Biotechnology,

Faculty of Advanced Medical Sciences, Tabriz University of Medical Sciences, Tabriz, Iran

Phone: +984133341933

Mobile: +989144059856

Alizadehe@tbzmed.ac.ir or e.alizadeh.2010@gmail.com

Response to respected reviewers/respected editors: 

1. The manuscript does not seem to have conducted any non-computational experiments (or computational experiments to analyze the cohort from the author's institution). At the same time, GSE15605 is a relatively small data sample (12+16), which makes the generalizability of the results may not be proven. The author should consider explaining this part or adding necessary data/experiments to provide necessary evidence for generalizability.

Answer: We greatly appreciate the reviewer‘s comment, a data which can be matched with our aims was very limited and considering the impact of this work and similar studies in the metastatic melanoma, we performed this study with that available size of data. Through a comprehensive analysis of the HPA database, we identified a set of hub genes that are overexpressed in metastatic melanoma. By leveraging the GEPIA database, we further validated these findings across multiple datasets, providing robust evidence for their association with this cancer type. GEPIA is a newly developed interactive web server for analyzing the RNA sequencing expression data of 9,736 tumors and 8,587 normal samples from the TCGA and the GTEx projects, using a standard processing pipeline. GEPIA provides customizable functions such as tumor/normal differential expression analysis, profiling according to cancer types or pathological stages, patient survival analysis, similar gene detection, correlation analysis and dimensionality reduction analysis.(Tang, Z. et al. (2017). Also, we included several references (25-28) to support the work.

2. Regarding the statement "The items recognized in the current study can be used as potential biomarkers for diagnostic, predictive, and might helpful to develop targeted combined therapies" mentioned in the manuscript. The experimental results do not seem to properly support the evidence mentioned in the manuscript. From my professional perspective, the manuscript seems to use virtual docking + molecular dynamics methods for analysis. As for molecular dynamics data, the relevant data of RMSF and SASA seem to be missing. As for the MMPBSA method, Figure 9 is very confusing, because such data may be more convincing in the form of a table. In addition, molecular dynamics does not seem to give a "conclusion". In addition, for example, Dabrafenib and MITF. In 2014, the FDA has approved Dabrafenib for use in patients with unresectable or metastatic melanoma with BRAF V600E mutations. To a certain extent, this coincides with the manuscript. In this context, is the addition of MITF a poly pharmacological effect? Or is it just a "computational coincidence"? More importantly, will it provide a proven potential predictor for more precise targeted therapy? Compared with a mechanical view of the compound, these issues should be discussed or anticipated by the author.

Answer: We appreciate the reviewer comment, the RMSF and SASA was performed and inserted in Methods, results (Figures: 9-12 and table 4) and in discussion (pages 9, 18, 19, and23).

Method 

Alterations in the structures of the yielded complexes were monitored through the root-mean-square deviation (RMSD), root mean square fluctuation (RMSF), and The Solvent Accessible Surface Area (SASA).

Result

The RMSF is primarily used to characterize and analyze local conformational changes within a protein structure. The Figure 9, the RMSF graphs of the STAT1-Guttiferone k, MITF-Dabrafenib, and ICAM1-Lifitegrast complexes are depicted in green, blue, and orange, respectively. Furthermore, the Fig.10 illustrates the SASA plot of all three complex structures within the simulation. The result shows that the MITF-Dabrafenib complex has higher sasa alteration than other two. The protein complexed with the selected ligands and their intermolecular interactions were analyzed using the Simulation Interactions Diagram (SID) during the 100 ns simulation run. These interactions (or "contacts") for the compounds with STAT1-Guttiferone k, MITF-Dabrafenib, and ICAM1-Lifitegrast complexes have been described based on hydrogen bonds, hydrophobic interactions, ionic interactions, and water bridges, as shown in Figure 11.

 However, for the MMPBSA method, replace Figure 9 with Table 4 for improved clarity.

In addition, molecular dynamics does not seem to give a "conclusion"

We added the Md simulation finalize to conclusion part.

The MD simulation outcomes indicated that all 3 complexes maintain stable in their interactions to the drug candidates, as illustrated in Figure. 8. Also, the RMSD graph shows that the MITF-Dabrafenib complex has lower value than others that indicates its more stability during simulation. The RMSF is primarily used to characterize and analyze local conformational changes within a protein structure. The Figure 9, the RMSF graphs of the STAT1-Guttiferone k, MITF-Dabrafenib, and ICAM1-Lifitegrast complexes are depicted in green, blue, and orange, respectively. Furthermore, the Fig.10 illustrates the SASA plot of all three complex structures within the simulation. The result shows that the MITF-Dabrafenib complex has higher SASA alteration than other two. The protein-ligand complexes were further examined for their MM/PBSA binding energy, as depicted in Table 4. Notably, the energy profiles of all complexes consistently displayed low values. Moreover, MITF-Dabrafenib complex shows lower energy than two other complexes. In next step, the protein complexed with the selected ligands and their intermolecular interactions were analyzed using the Simulation Interactions Diagram (SID) during the 100 ns simulation run. These interactions (or "contacts") for the compounds with STAT1-Guttiferone k, MITF-Dabrafenib, and ICAM1-Lifitegrast complexes have been described based on hydrogen bonds, hydrophobic interactions, ionic interactions, and water bridges, as shown in Figure 11. The summary of MD simulation results show that all three drug protein complexes are stable during simulation and can be suggest for future work, however, among them MITF-Dabrafenib complex is more stable.

3. I personally suggest that the manuscript should add a graphic Abstract to attract readers. And many of the manuscript's expression systems face major revisions (for example, some pictures are marked as Fig, and some are Figure).

Answer: Thanks to respected reviewer, graphical abstract was added to revised version because the journal has no graphical abstract in its format/ style we put it as figure 1 (page 5) and following figures were renumbered. The word Fig was replaced with Figure in all manuscript of revised version.

https://doi.org/10.1007/s12079-021-00612-8

Answer: Thanks a lot for comment, the reference was inserted in revised version page 11.line 10 Ref no. 25. Also, the whole manuscript double checked by ithenticate plagiarism checker and the probable overlapping were rewritten.

4. Funding Information and Financial Disclosure sections do not match: We note that the grant information you provided in the ‘Funding Information’ and ‘Financial Disclosure’ sections do not match. When you resubmit, please ensure that you provide the correct grant numbers for the awards you received for your study in the ‘Funding Information’ section.

Answer: Thanks a lot for comment, the funding information was corrected in revised version (page 27).

[None]. 

Answer: Thanks a lot for comment, there is no conflict of interest and the statement recommended inserted in revised version (page 27).

6. Thank you for stating the following financial disclosure: 

 [The authors would like to acknowledge the Department of Medical Biotechnology, Faculty of Advanced Medical Sciences, Tabriz University of Medical Sciences for all support provided [69755].]. 

Answer: Thanks a lot for comment, the recommended text was included in funding section (page 27).

7. Thank you for stating the following in the Acknowledgments Section of your manuscript: 

[The authors would like to acknowledge the Department of Medical Biotechnology, Faculty of Advanced Medical Sciences, Tabriz University of Medical Sciences for all support provided [69755]],We note that you have provided funding information that is not currently declared in your Funding Statement. However, funding information should not appear in the Acknowledgments section or other areas of your manuscript. We will only publish funding information present in the Funding Statement section of the online submission form. 

 [The authors would like to acknowledge the Department of Medical Biotechnology, Faculty of Advanced Medical Sciences, Tabriz University of Medical Sciences for all support provided [69755].]. Please include your amended statements within your cover letter; we will change the online submission form on your behalf.

Answer: Thanks a lot for comment, the requested explanations were included in M&M section page 5 in revised version (page 6).

8. In the online submission form, you indicated that [All data will be presented upon request.]. 

Answer: Thanks a lot for comment, the Data availability was changed to “ all data are available within the manuscript itself (page 27). 

9. Your ethics statement should only appear in the Methods section of your manuscript. If your ethics statement is written in any section besides the Methods, please move it to the Methods section and delete it from any other section. Please ensure that your ethics statement is included in your manuscript, as the ethics statement entered into the online submission form will not be published alongside your manuscript. 

Answer: Thanks a lot for comment, the ethics statement was inserted in M&M and removed from Acknowledgment section, (page 5 in revised version).

10. We note that Figure(s) 3a in your submission contain copyrighted images. All PLOS content is published under the Creative Commons Attribution License (CC BY 4.0), which means that the manuscript, images, and Supporting Information files will be freely available online, and any third party is permitted to access, download, copy, distribute, and use these materials in any way, even commercially, with proper attribution. For more information, see our copyright guidelines: http://journals.plos.org/plosone/s/licenses-and-copyright.

. You may seek permission from the original copyright holder of Figure(s) 3a to publish the content specifically under the CC BY 4.0 license. 

b If you are unable to obtain permission from the original copyright holder to publish these figures under the CC BY 4.0 license or if the copyright holder’s requirements are incompatible with the CC BY 4.0 license, please either i) remove the figure or ii) supply a replacement figure that complies with the CC BY 4.0 license. Please check copyright information on all replacement figures and update the figure caption with source information. If applicable, please specify in the figure caption text when a figure is similar but not identical to the original image and is therefore for illustrative purposes only.

Answer: Thanks a lot for comment, 

The dataset analyzed in the current study is available in the [GSE datasets and Human Protein Atlas] repository, [https://www.ncbi.nlm.nih.gov/geo/query/acc.cgi?acc=GSE15605] and [https://www.proteinatlas.org/ accession numbers, ENSG00000169248, ENSG00000090339, ENSG00000138795, ENSG00000187098 and ENSG00000115415].

All data generated during this study are included in this published article, please check Data availability. Because we used the images from Human protein atlas figure in figure3a in revised version figure 4a, the figures are freely available as stated in this link: https://www.proteinatlas.org/about/licence and https://v18.proteinatlas.org/about/licence. Therefore, we just cited the Human protein atlas in figure 4a legend (page 35). 

11. Please include a copy of Table 1, 2 and 3 which you refer to in your text on page 11 and 13.

Answer: Thanks a lot for comment, all tables were included in revised version 

---

## [Editor Report · Decision Letter 1]

14 Oct 2024

Metastatic melanoma: An integrated analysis to identify critical regulators associated with prognosis, pathogenesis and targeted therapies

PONE-D-24-19761R1

Dear Dr. Alizadeh,

We’re pleased to inform you that your manuscript has been judged scientifically suitable for publication and will be formally accepted for publication once it meets all outstanding technical requirements.

Kind regards,

Ruo Wang

Academic Editor

PLOS ONE
---

## [Editor Report · Acceptance letter]

3 Dec 2024

PONE-D-24-19761R1 

PLOS ONE

Dear Dr. Alizadeh, 

I'm pleased to inform you that your manuscript has been deemed suitable for publication in PLOS ONE. Congratulations! Your manuscript is now being handed over to our production team.

Kind regards, 

on behalf of

Dr. Ruo Wang 

Academic Editor

PLOS ONE